# Enhancing carbohydrate repartitioning into lipid and carotenoid by disruption of microalgae starch debranching enzyme

Yuichi Kato [1], Tomoki Oyama[2], Kentaro Inokuma[1], Christopher J. Vavricka [2], Mami Matsuda[2], Ryota Hidese[2], Katsuya Satoh[3], Yutaka Oono [3], Jo-Shu Chang[4,5,6], Tomohisa Hasunuma [1,2✉] & Akihiko Kondo [1,2,7]

Light/dark cycling is an inherent condition of outdoor microalgae cultivation, but is often unfavorable for lipid accumulation. This study aims to identify promising targets for metabolic engineering of improved lipid accumulation under outdoor conditions. Consequently, the lipid-rich mutant *Chlamydomonas* sp. KOR1 was developed through light/dark-conditioned screening. During dark periods with depressed $CO_2$ fixation, KOR1 shows rapid carbohydrate degradation together with increased lipid and carotenoid contents. KOR1 was subsequently characterized with extensive mutation of the *ISA1* gene encoding a starch debranching enzyme (DBE). Dynamic time-course profiling and metabolomics reveal dramatic changes in KOR1 metabolism throughout light/dark cycles. During light periods, increased flux from $CO_2$ through glycolytic intermediates is directly observed to accompany enhanced formation of small starch-like particles, which are then efficiently repartitioned in the next dark cycle. This study demonstrates that disruption of DBE can improve biofuel production under light/dark conditions, through accelerated carbohydrate repartitioning into lipid and carotenoid.

[1] Engineering Biology Research Center, Kobe University, Kobe, Japan. [2] Graduate School of Science, Technology and Innovation, Kobe University, Kobe, Japan. [3] Project "Ion Beam Mutagenesis", Department of Radiation-Applied Biology Research, Takasaki Advanced Radiation Research Institute, Quantum Beam Science Research Directorate, National Institutes for Quantum and Radiological Science and Technology, Takasaki, Gunma, Japan. [4] Department of Chemical and Materials Engineering, Tunghai University, Taichung, Taiwan. [5] Research Center for Smart Sustainable Circular Economy, Tunghai University, Taichung, Taiwan. [6] Department of Chemical Engineering, National Cheng Kung University, Tainan, Taiwan. [7] Department of Chemical Science and Engineering, Graduate School of Engineering, Kobe University, Kobe, Japan. ✉email: hasunuma@port.kobe-u.ac.jp

Microalgae typically partition carbon resources derived from $CO_2$ into energy storage compounds such as starch and lipid under adverse environments including nutrient depletion and salinity stress[1–3]. Due to their high potential to photosynthetically produce lipid from $CO_2$ in the atmosphere, microalgae have been widely studied as a promising next-generation biofuel producer[4,5]. Photoautotrophic cultivation using sunlight as the energy source is essential for cost-effective production of microalgal biofuels, and therefore, diurnal fluctuation of light intensity (i.e. light/dark cycling) is an assumed condition.

Light/dark cycling is an important factor for phototrophic organisms including microalgae. In the model green microalga *Chlamydomonas reinhardtii*, genome-wide gene expression of central carbon metabolism is regulated under diurnal periodicity, with starch content increasing during light periods and peaking in the middle of dark periods[6,7]. Other cell compositions such as lipid content are also reported to be influenced by light/dark cycling. Unfortunately, lipid accumulation decreases under light/dark cycling in lipid-rich microalgae such as *Dunaliella viridis*, *Tribonema minus*, and *Chlamydomonas* sp., relative to that of continuous illumination[8–10]. For successful commercialization, lipid accumulation under light/dark conditions must be improved by metabolic engineering approaches, such as mutational breeding and genetic engineering[11,12].

Mutational breeding of lipid-rich mutant microalgae has already been performed in previous studies[13–15]. Fluorescence-activated cell sorting (FACS) is a powerful tool for accelerating selective breeding of lipid-rich mutants. With application of lipophilic fluorescent dyes such as Nile Red and boron-dipyrromethene (BODIPY)[16], FACS enables acquisition of cells with higher lipid accumulation in a high-throughput manner. Lipid-rich mutants in diverse microalgal species, including *C. reinhardtii, Chlorococcum littorale*, and *Euglena gracilis*, were previously obtained by the selective breeding approach using FACS[14,15,17–19]. However, these screenings have only been conducted under the optimal conditions for lipid accumulation (for example, continuous illumination), and did not simulate unfavorable environmental factors that will be faced in outdoor cultivation (for example, light/dark cycling). Accordingly, metabolic engineering approaches for overcoming decreased lipid accumulation under light/dark cycling have not been reported yet.

The current study aims to identify a hopeful target for metabolic engineering to improve microalgal lipid accumulation under light/dark conditions. Here, the green microalga *Chlamydomonas* sp. JSC4 was selected as the parent strain for mutational breeding under light/dark cycling. JSC4 accumulates lipid under nitrogen-deplete and salinity stress conditions, and achieves high lipid content together with high biomass production[20,21]. These beneficial features of JSC4 are related to starch-to-lipid biosynthesis switching mechanisms[22], however, lipid content of JSC4 significantly decreases under light/dark conditions, compared to that of continuous illumination[10].

Here, mutational breeding resulted in a novel lipid-rich mutant KOR1, by way of combined carbon beam irradiation and light/dark-conditioned FACS screening. This mutant shows higher lipid content relative to that of the parent strain JSC4 under light/dark conditions. To elucidate the metabolic mechanism underlying enhanced lipid accumulation in KOR1, detailed time-course profiling of cell components together with metabolome analysis were performed under light/dark conditions. Insertion/deletion mutations in an isoamylase-type starch debranching enzyme (DBE) gene *ISA1* were determined in KOR1, and found to accompany augmented carbohydrate metabolism and lipid accumulation. These experiments demonstrate that disruption of DBE enhances carbohydrate degradation and repartitioning of

carbon resources into lipid and carotenoid. This establishes DBE as a hopeful metabolic engineering target to improve lipid accumulation under light/dark cycling, which is a representative natural environmental condition.

## Results

**Selective breeding of lipid-rich mutant KOR1.** Lipid content of microalgae under natural light/dark cycling must be further improved to meet biofuel demands[11]. To identify a promising metabolic engineering target that can be modified to improve lipid content, lipid-rich mutants were first screened under light/dark conditions. *Chlamydomonas* sp. JSC4 was selected as the parental strain for mutagenesis and light/dark-conditioned screening because lipid accumulation significantly decreases under the conditions in this microalga[10]. Mutagenized cells derived from JSC4 were generated with carbon beam irradiation[23,24], and cultured under light/dark cycling conditions. Then, lipid-rich cells that presented relatively higher BODIPY fluorescence were sorted by way of FACS. The secondary screening was performed by measuring lipid content of individual candidate mutant strains in the small-scale cultivation under light/dark conditions. Consequently, the mutant strain KOR1, which showed 2.28-fold increase in lipid content relative to JSC4 in the secondary screening, was identified (Supplementary Fig. 1).

**Increase in lipid along with significant decrease in carbohydrate under dark conditions in KOR1.** To evaluate the lipid production in detail, time-course analysis of nitrate consumption, biomass, lipid content, and carbohydrate content was performed in KOR1 grown under light/dark conditions. In the parental strain JSC4, lipid accumulation and starch degradation occur after nitrogen depletion[10]. In KOR1, consumption of nitrogen is similar to that in JSC4: nitrate is mainly consumed during the light periods in both strains at the same rate and completely depleted by day 4.0 (Fig. 1a). During the nitrate replete condition before day 4.0, KOR1 biomass is similar to that of JSC4, while under the nitrate deplete condition after day 4.0, the mutant biomass is lower than that of the parental strain (Fig. 1b). Differences in lipid content per dry cell weight (DCW) between these strains appear only after nitrate depletion, and lipid accumulation of KOR1 accelerates primarily during the dark periods (Fig. 1c). Lipid content of JSC4 and KOR1 at day 12.0 is 21.0% and 42.2%, respectively. This 2.01-fold increase in lipid content of KOR1 is accompanied by a striking fluctuation in carbohydrate content in response to light/dark cycles (Fig. 1d). Within this cycle, KOR1 carbohydrate per DCW content increases to a maximum of 22.5% under light and decreased to almost 0.0% under darkness. In the dark phase, KOR1 production of high lipid content occurs together with rapid consumption of carbohydrate content in addition to the natural reduction of $CO_2$ fixation. The significant decrease in carbohydrate content during dark periods must be due to enhanced carbohydrate degradation in KOR1. Therefore, it was hypothesized that the increase in lipid content under dark conditions is due to repartitioning of intracellular carbon resources generated from the degradation of carbohydrate.

**Significant mutations of an isoamylase-type DBE gene *ISA1* found in KOR1.** The highly polarized carbohydrate cycling of KOR1 (Fig. 1d) strongly suggested a mutation related to starch synthesis/degradation. This is consistent with previous reports that diverse microalgae with impaired starch synthesis show increased lipid accumulation[25–31], although the striking fluctuation in carbohydrate content is firstly reported in this study. To pinpoint the genetic variations in KOR1, the whole genome sequence of KOR1 was determined and compared to that of

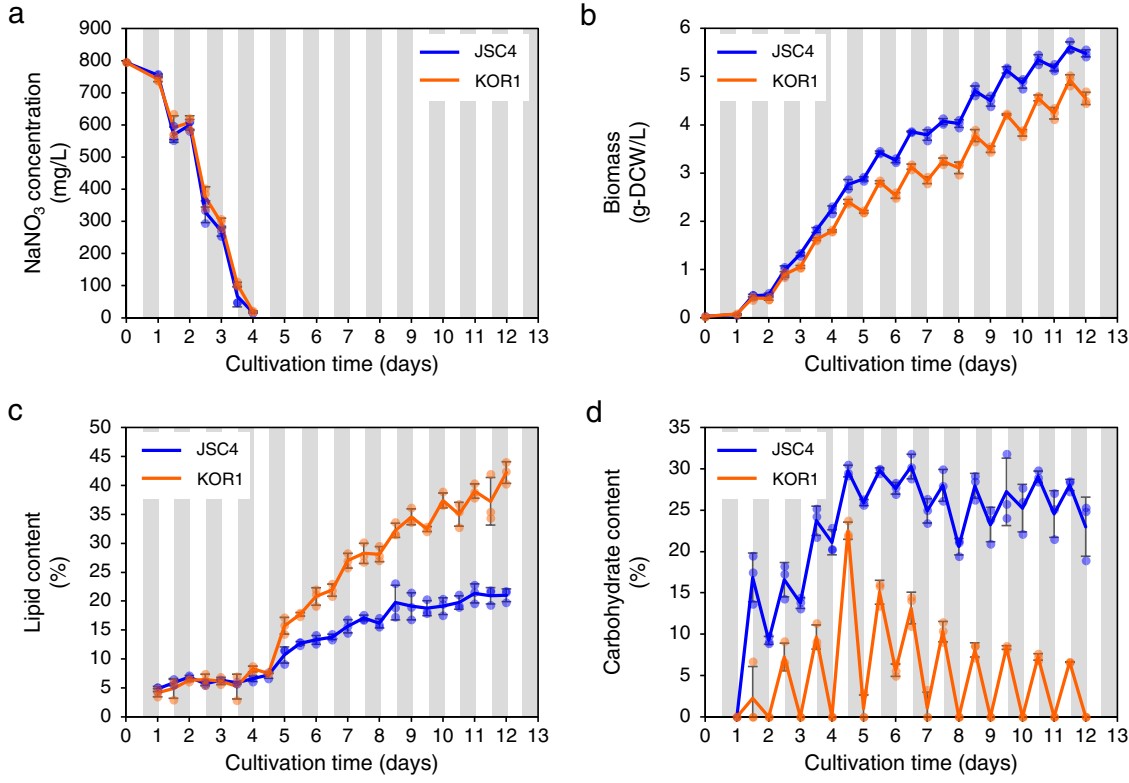

**Fig. 1 Time-course profiles of nitrogen consumption and energy storage. a** Residual nitrate concentration in the culture media. **b** Biomass. **c** Lipid content. **d** Carbohydrate content. White and gray bands represent light and dark periods, respectively. Error bars indicate the standard deviation of three replicate experiments.

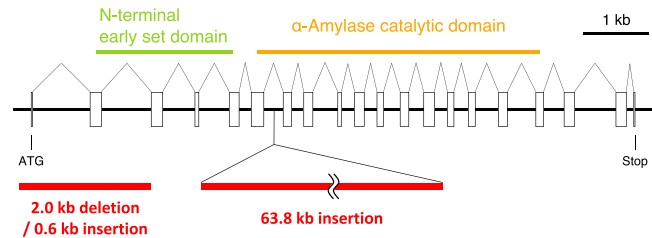

**Fig. 2 Mutations in the *ISA1* gene of *Chlamydomonas* sp. KOR1.**
*Chlamydomonas* sp. ISA1 encodes an isoamylase-type starch debranching enzyme (DBE), with an N-terminal early set domain and C-terminal α-amylase catalytic domain. Predicted exons are shown as white rectangles. Deletion and insertion mutation sites in KOR1 ISA1 are shown as red bars.

JSC4[22]. The *Chlamydomonas* sp. used in this study possesses five soluble starch synthase genes (EC: 2.4.1.21), 2 granule bound starch synthase genes (EC: 2.4.1.242), 3 starch branching enzyme genes (EC: 2.4.1.18), 4 starch phosphorylase genes (EC: 2.4.1.1), 3 isoamylase-type DBE genes (EC: 3.2.1.68), 2 α-amylase genes (EC: 3.2.1.1), and 3 β-amylase genes (EC: 3.2.1.2), all related to starch synthesis/degradation. Within these genes, it is noteworthy that exon mutations are only confirmed in the *ISA1* gene in the KOR1 genome (Fig. 2). The *ISA1* gene in *Chlamydomonas* sp. encodes an isoamylase-type DBE, with an N-terminal early set domain (cd02856) and C-terminal α-amylase catalytic domain (cd11326). In the *ISA* gene of KOR1, a 2.0 kb sequence covering the initiation codon through part of the N-terminal early set domain is deleted and substituted by a 0.6 kb sequence. Also, a 63.8 kb sequence consisting of the deleted sequence above and at least six fragments from the other genomic regions are inserted into the α-amylase catalytic domain. The large deletion and insertion mutations strongly suggest that *ISA1* does not function in KOR1. Loss of

DBE activity in KOR1 was validated by zymography[32], where significantly depressed activity toward amylopectin compared to that of JSC4 is observed in KOR1 (Supplementary Fig. 2). In addition, to confirm that DEB deficiency is responsible for KOR1 phenotypes, the *C. reinhardtii sta7-10* mutant, a DBE-deficient mutant isolated in a previous study[33], was also characterized. Similar to KOR1, the *sta7-10* mutant exhibits a significant increase in lipid content and fluctuation in carbohydrate content during light/dark cycling (Supplementary Fig. 3). Therefore, DBE appears as the prime target to enhance repartitioning of carbohydrate into lipid under light/dark conditions.

**Accumulation of starch-like small granules in KOR1 cells.** In plants and microalgae, loss of DBE activity is reported to cause structural changes in starch, resulting in highly branched and water-soluble polysaccharide, called phytoglycogen[34–37]. Functional deficiency of *ISA1* suggested accumulation of carbohydrate as phytoglycogen as a potential KOR1 phenotype. Thus, the cellular structure of KOR1 carbohydrate was observed by transmission electron microscopy (TEM). Under the nitrate replete condition of day 1.5, starch granule formation around the pyrenoid is observed in JSC4 cells (Fig. 3). Under the nitrate deplete condition, both lipid droplets and starch granules are found in JSC4 cells regardless of dawn (day 10.0) and dusk (day 10.5). In contrast, KOR1 cells contain many small particles formed around the pyrenoid in place of any large starch granules, under the nitrate replete condition at day 1.5. These small particles are also found under the nitrate deplete condition, with fewer appearing at the end of dark periods (day 10.0). In addition to the location and timing of appearance with carbohydrate content increase, the high electron density after lead staining indicates that these small particles are structurally abnormal starch, and likely a type of phytoglycogen[38]. A reduction in number of KOR1

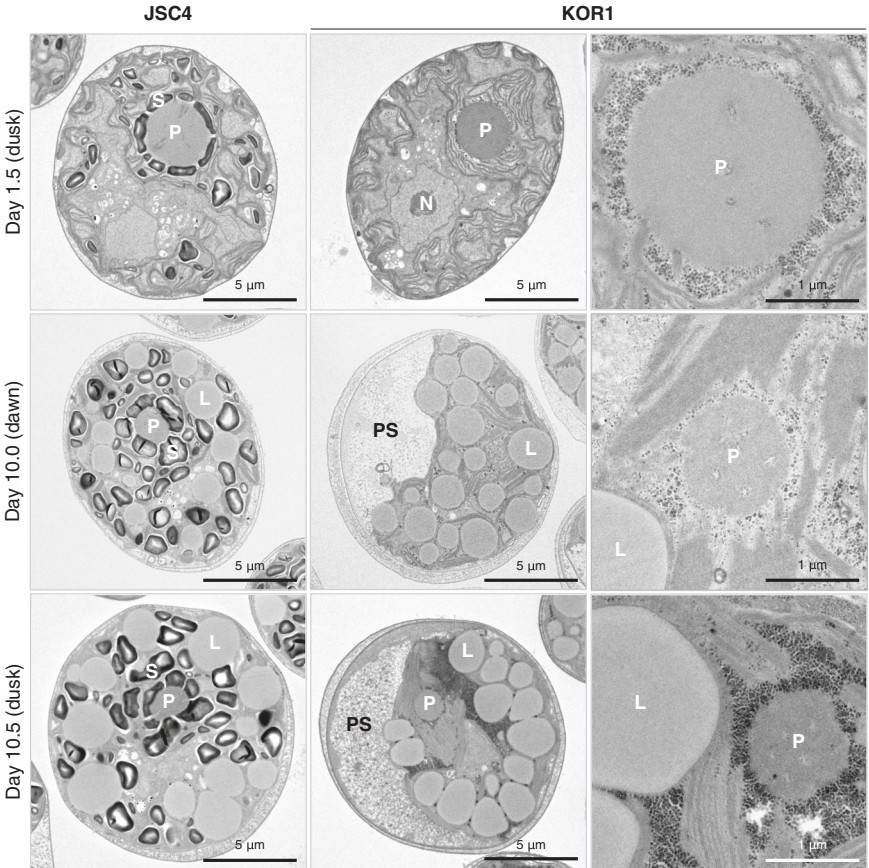

**Fig. 3 Cell morphology viewed by transmission electron microscopy (TEM).** Cells were harvested under nitrate-replete conditions (Day 1.5) and nitrate-deplete conditions (Day 10.0 and Day 10.5), fixed immediately, and visualized. Day 1.5 and Day 10.5 are just before the end of a light period (dusk) and Day 10.0 is just before the end of a dark period (dawn). S: starch granule, L: lipid droplet, P: pyrenoid, N: nucleus, PS: periplasmic space.

granules accompanied accelerated carbohydrate degradation under dark conditions (Fig. 1d), indicating that these starch-like small granules are highly degradable. In addition, a large periplasmic space is observed in KOR1 cells under the nitrate deplete conditions (day 10.0 and 10.5), which may be explained by the loss of large starch granules.

**Accumulation of lutein and β-carotene in KOR1.** Since lipophilic carotenoids can accumulate in lipid droplets[39,40], the influence of *ISA1* deficiency on pigment accumulation was investigated. JSC4 contains abundant carotenoids, especially lutein[41,42]. Lutein content of KOR1 was similar to that of JSC4 during the nitrate replete condition, but was significantly higher under the nitrate deplete condition after day 4.0 (Fig. 4a). The maximal lutein contents of JSC4 and KOR1 are 2.08 mg g-DCW$^{-1}$ (day 7.0) and 2.64 mg g-DCW$^{-1}$ (day 8.0), respectively. β-Carotene, another major carotenoid in green microalgae including *Chlamydomonas*, also increases in KOR1 under nitrate deplete conditions (Fig. 4b). Maximal β-carotene content of JSC4 and KOR1 are 1.00 mg g-DCW$^{-1}$ (day 2.5) and 1.27 mg g-DCW$^{-1}$ (day 5.0), respectively. DCW-based contents of KOR1 lutein and β-carotene fluctuate in response to light/dark cycling with decreases during the light periods and increases during the dark periods, suggesting a relationship to lipid and carbohydrate accumulation. Chlorophyll $a + b$ content does not dramatically change in KOR1, although there is a temporal increase during dark periods under nitrate deplete conditions (Fig. 4c). Similar changes in pigments are observed in the *C. reinhardtii sta7-10* mutant (Supplementary Fig. 4), further supporting that pigment phenotypes in KOR1 are also caused by DBE deficiency.

**Comprehensive increase of intermediate metabolites in KOR1.** In order to elucidate the metabolic mechanism underlying the increased KOR1 lipid and carotenoid content, diurnal time-course profiling of intermediate metabolites was performed during the nitrate deplete condition from day 5.0 to day 6.0 (Fig. 5). This experiment shows a dramatic increase in KOR1 carbohydrate from 2.1% to 19.6% during the light period, followed by sharp a decrease back to 2.1% during the dark period. Lipid content increases from 12.2% to 16.9% during the dark period in KOR1, but the corresponding period is marked by no change in JSC4. Lutein content in KOR1 temporary increases from 2.04 to 2.62 mg g-DCW$^{-1}$ during the light period at zeitgeber time (ZT) = 3 h, and also increases from 2.20 to 2.61 mg g-DCW$^{-1}$ during the dark period.

Synthetic pathways in *Chlamydomonas* sp. to produce starch and lipid from $CO_2$ have been previously indicated[22]. In this study, the metabolites in the Calvin cycle including ribulose-1,5-bisphosphate (RuBP), 3-phosphoglycerate (3PG), erythrose 4-phosphate (E4P), sedoheputulose 7-phosphate (S7P), and ribulose 5-phosphate (Ru5P) significantly increase in KOR1 after the initiation of the light period. In particular, the level of 3PG increases at ZT = 3 h and remains higher during the light period in KOR1, while 3PG shows no significant change during the light and dark period in JSC4. In KOR1, increased levels of downstream metabolites are also detected during the light period, including intermediates of starch, lipid, and carotenoid pathways. In the starch synthesis/degradation pathway, levels of fructose-6-phosphate (F6P), glucose-6-phosphate (G6P), and ADP-glucose comprehensively increase in KOR1. In the pathway for lipid synthesis, levels of phosphoenolpyruvate (PEP), pyruvate,

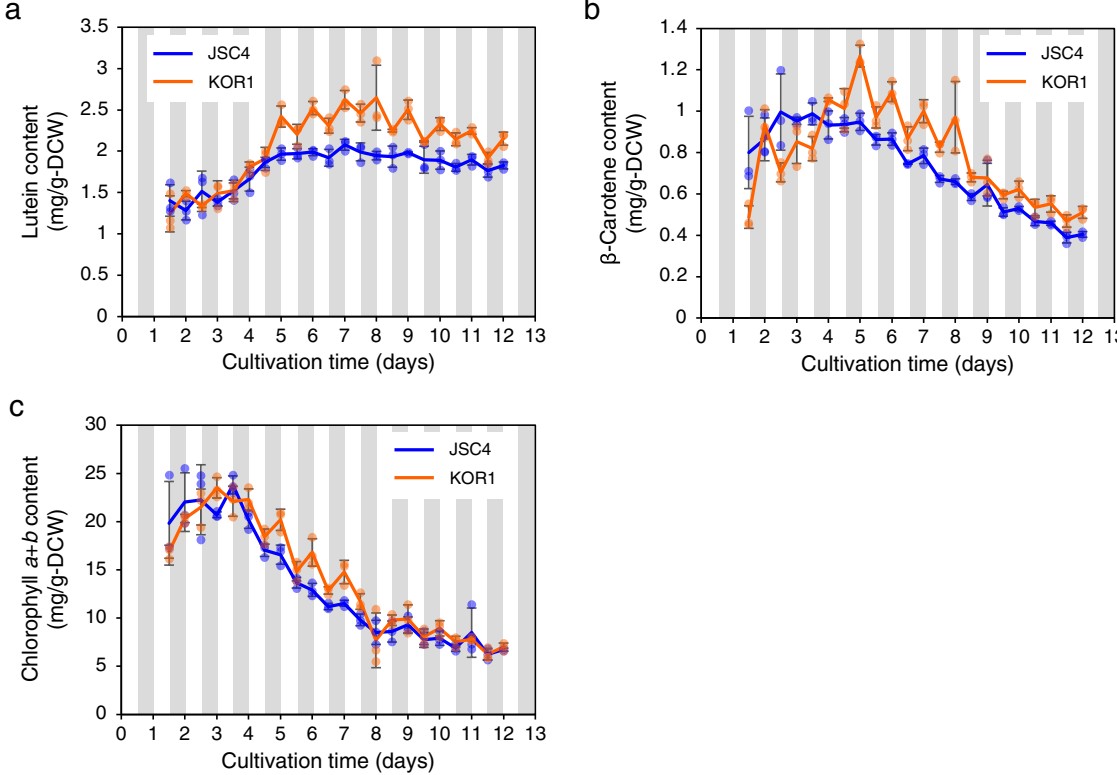

**Fig. 4 Time-course profiles of photosynthetic pigments. a** Lutein content. **b** β-Carotene content. **c** Chlorophyll $a+b$ content. White and gray bands represent light and dark periods, respectively. Error bars indicate the standard deviation of three replicate experiments.

acetyl-CoA, and glycerol 3-phosphate (G3P) increase in KOR1, mainly during the light period. In the non-mevalonate pathway (MEP pathway) which is upstream of the carotenoid synthesis pathway, levels of 1-deoxy-D-xylulose 5-phosphate (DXP) and 2-C-methyl-D-erythritol-2,4-cyclopyrophosphate (MEcPP) also increase in KOR1, mainly during the light period. Thus, many metabolites throughout central carbon metabolism show significant increases in KOR1, particularly during light periods. Increased levels of KOR1 intermediate metabolites indicate that abundant metabolites are available as the substrates of enhanced lipid and carotenoid synthesis.

**Increased de novo synthesis of intermediate metabolites from $CO_2$ in KOR1.** In KOR1, enhanced carbohydrate degradation and repartitioning of the carbon resource into lipid during dark periods is suggested by the results above. However, without dynamic metabolomics, it is unclear whether or not carbon flux from $CO_2$ is enhanced during the light periods in KOR1. To directly observe central metabolic flux from $CO_2$, in vivo [13]C labeling experiment was performed, focusing on targeted metabolites in the Calvin cycle through the modified pathways of starch and lipid[10,22,43]. To do this, cultured cells were harvested at day 5.5, resuspended in medium containing $NaH^{13}CO_3$ (labeling time = 0 min), incubated under illumination, and chronologically sampled at 2.5, 5, and 10 min. The level of newly synthesized metabolites was evaluated based on the fraction of [13]C-labeled metabolites (referred as "[13]C labeling" in this study) which is calculated by multiplying metabolic pool size by the [13]C labeled ratio. [13]C labeling of 3PG, a major metabolite in the Calvin cycle, significantly increases in KOR1 (Fig. 6). Within the starch synthesis/degradation pathway, [13]C labeling of F6P and G6P also significantly increases in KOR1. In the lipid synthesis pathway,

[13]C labeling of PEP and G3P increase in KOR1 while labeling of pyruvate and acetyl-CoA are not statistically significant. Thus, de novo synthesis of key metabolites involved in starch synthesis and in a part of lipid synthesis is enhanced in KOR1 under the illuminated condition.

**Discussion**
This study provides an insight that disruption of the DBE gene is a prime approach to improve lipid accumulation under light/dark conditions. Several studies previously performed selective breeding of lipid-rich mutant microalgae using FACS[14,15,17–19]. However, the screenings of the previous studies have been conducted only under continuously illuminated conditions, and moreover, neither causative genes nor fundamental metabolic mechanisms for increased lipid accumulation were reported. To obtain a mutant microalga in which high lipid accumulation is ensured under natural light cycling, this study preformed screening under light/dark conditions. In addition, a potential causative gene and fundamental metabolic mechanism are identified in a lipid-rich mutant, obtained by ion beam mutagenesis and FACS-based screening. Analysis of the lipid-rich mutant KOR1 indicates that DBE deficiency enhances the degradation of carbohydrates for repartitioning of carbon resources into lipid/carotenoid, and not just a redirection of carbon flux toward lipid synthesis[25,44]. The carbon partition/repartition model in DBE-deficient microalgae proposed from this study is shown in Fig. 7. Under light conditions, atmospheric $CO_2$ is fixed in the Calvin cycle, and the carbon resources are primarily captured as water-soluble phytoglycogen instead of insoluble starch. Under dark conditions, accumulated phytoglycogen can be more easily degraded and converted into intermediate metabolites, and then used as the substrate for synthesizing lipid and carotenoid.

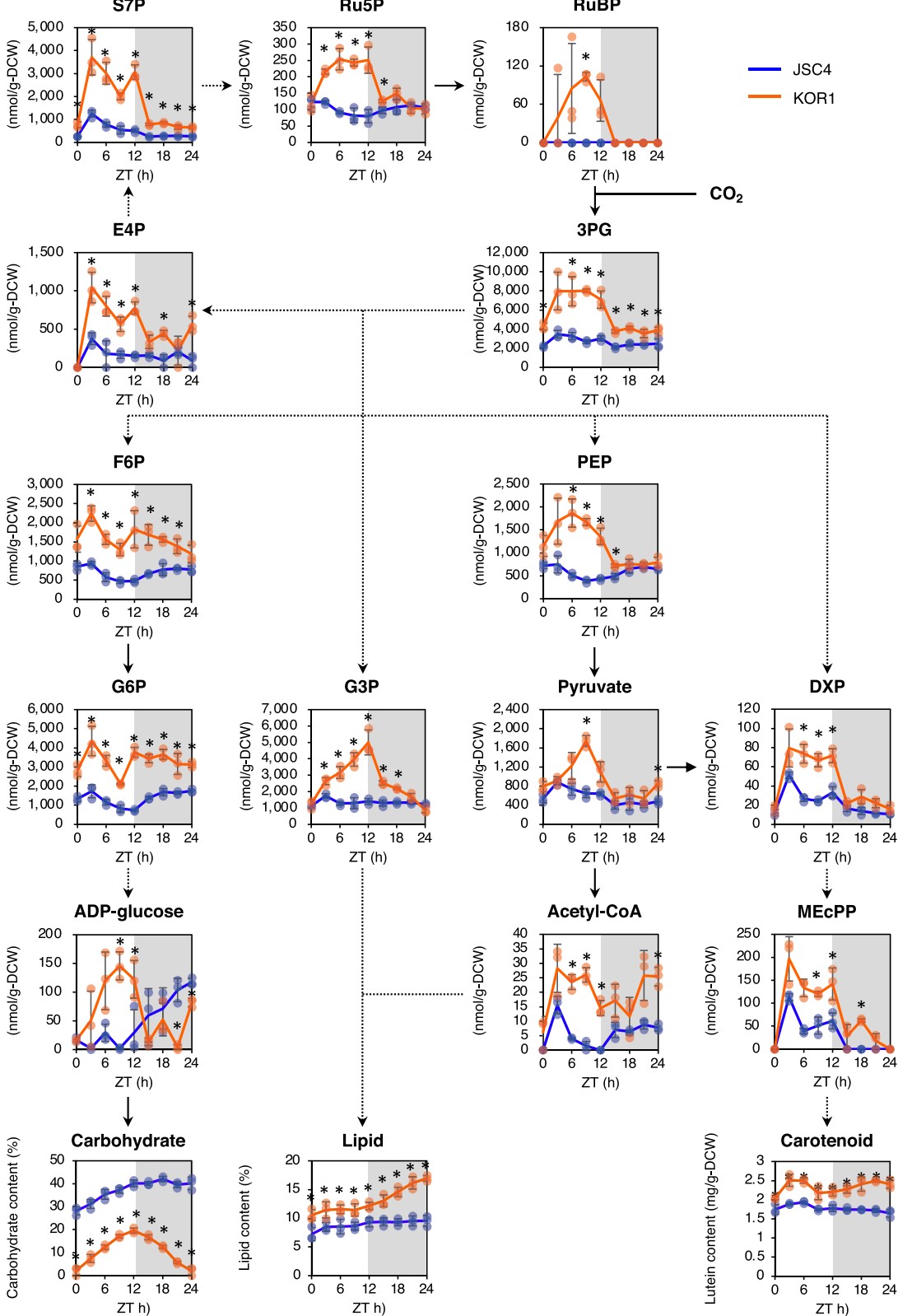

**Fig. 5 Diurnal profiles of the metabolic pool size.** ZT zeitgeber time, S7P sedoheptulose 7-phosphate, Ru5P ribulose 5-phosphate, RuBP ribulose-1,5-bisphosphate, 3PG 3-phosphoglycerate, E4P erythrose 4-phosphate, F6P fructose 6-phosphate, G6P glucose 6-phosphate, G3P glycerol 3-phosphate, PEP phosphoenolpyruvate, DXP 1-deoxy-D-xylulose 5-phosphate, MEcPP 2-C-methyl-D-erythritol-2,4-cyclopyrophosphate. Reactions containing single and multiple enzymatic steps are represented by solid and dotted lines, respectively. White and gray bands represent light and dark periods, respectively. Error bars indicate the standard deviation of three replicate experiments (*$p < 0.05$ by Welch's $t$ test).

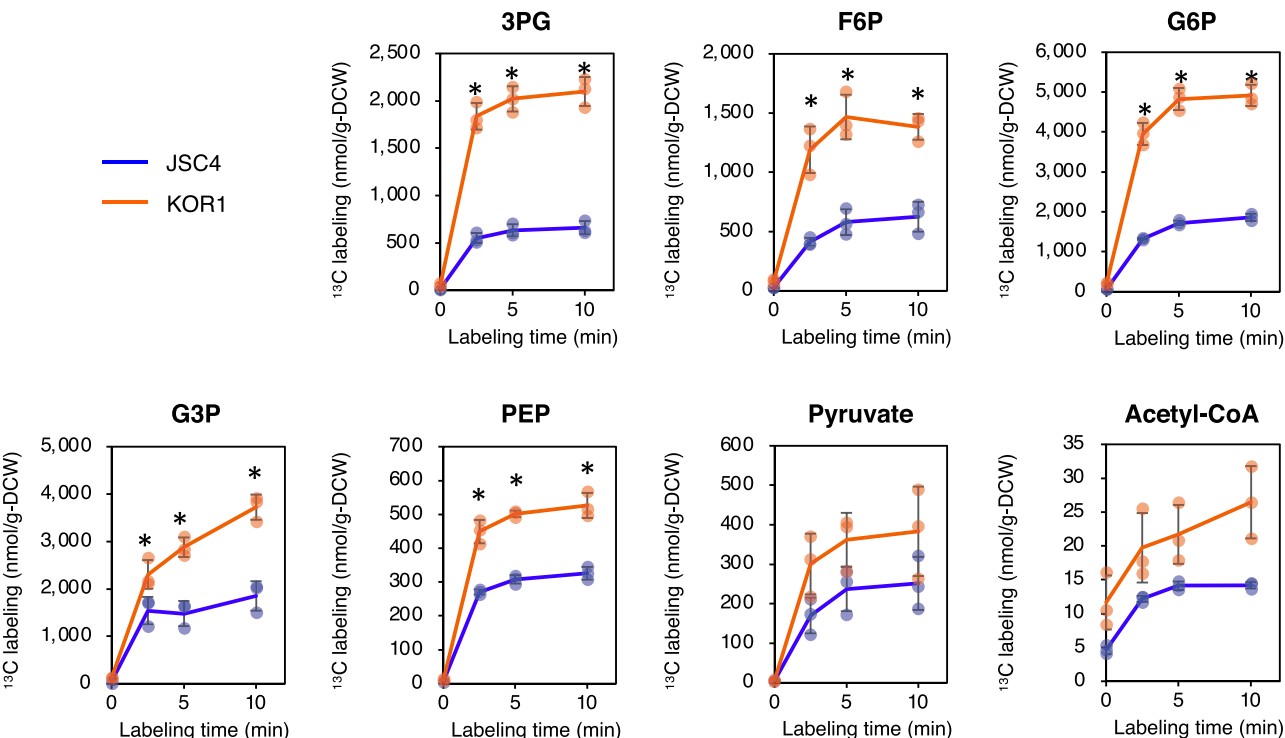

**Fig. 6 Dynamic metabolic profiles under illuminated conditions.** Cells grown under the light/dark conditions for 5.5 days were harvested and resuspended in medium containing $NaH^{13}CO_3$. Vertical axes: level of $^{13}C$-labeled metabolites calculated by multiplying metabolic pool size with $^{13}C$-labeled ratio. Horizontal axes: labeling time after starting $^{13}C$ supply to the cells. Error bars indicate the standard deviation of three replicate experiments (*$p < 0.05$ by Welch's $t$ test).

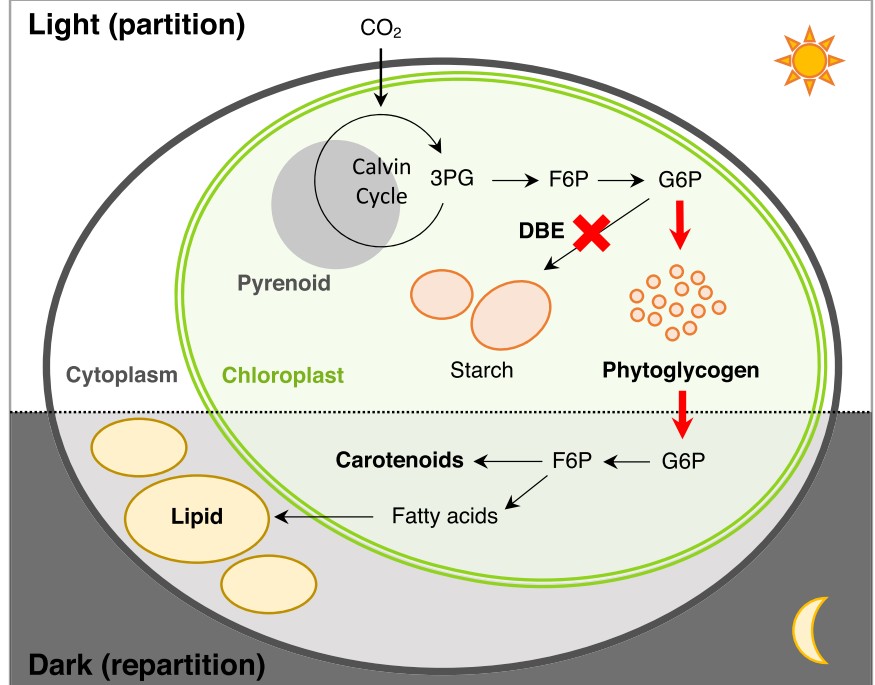

**Fig. 7 Carbon partition/repartition model for DBE-deficient microalgae under light/dark conditions.** DBE-deficient microalgae accumulate carbohydrate as phytoglycogen, instead of starch. Under light conditions, carbon resources derived from $CO_2$ are partitioned into phytoglycogen. Under dark conditions, phytoglycogen is rapidly converted into intermediate metabolites and then repartitioned into lipid/carotenoid.

The maximal lipid productivities of JSC4 and KOR1 are 109.3 mg $L^{-1}$ $day^{-1}$ (day 8.5) and 160.0 mg $L^{-1}$ $day^{-1}$ (day 12.0), respectively. Thus, breeding of KOR1 resulted in a 1.46-fold improvement in lipid productivity under light/dark conditions which is necessary for outdoor cultivation. For further improvement of lipid productivity, culture conditions should be optimized, for example, by applying a salinity-gradient strategy[21,41]. Previous studies have clarified that impaired starch synthesis typically causes increased lipid accumulation in the diverse microalgae, including *C. reinhardtii*, *Chlorella pyrenoidosa*, *Dunaliella tertiolecta* and *Scenedesmus obliquus*[25–31]. For example, a *C. reinhardtii* mutant of *sta6*, encoding ADP-glucose pyrophosphorylase (AGPase), was unable to convert glucose-1-phosphate (G1P) to the starch precursor ADP-glucose[45], while showing 10-fold increase in lipid content[25–27,46]. Another *C. reinhardtii* mutant *sta7-10*, deficient in an isoamylase-type DBE, showed a 4-fold increase in lipid content[28]. The light/dark cycling response of the *sta7-10* mutant, which has much in common to KOR1, is reported in this study (Supplementary Figs. 3 and 4). Although redirection of carbon flux toward lipid synthesis was assumed as a possible metabolic basis for the lipid accumulation[25,44], the fundamental metabolic mechanisms for lipid accumulation have not been clarified in these initial studies. A significant decrease in KOR1 carbohydrate (Fig. 1d) suggests that carbon resources generated from carbohydrate degradation are used for lipid synthesis. The increase in lipid content is also observed under continuous illumination (Supplementary Fig. 5), indicating that dark periods are not essential for lipid accumulation in DBE-deficient microalgae.

This study determined large insertion/deletion mutations in *ISA1* as causative mutations in KOR1 (Fig. 2). Highly depressed DBE activity in KOR1 suggests that ISA1 is the major DBE of the *Chlamydomonas* sp. used in this study (Supplementary Fig. 2). Since signals of the DBE activity were detected at the top of the gel, the *Chlamydomonas* sp. ISA1 might function as complex similar to the other microalgae and plants. Within *C. reinhardtii*, catalytically active ISA1 functions as a homodimer or heteromeric complex with catalytically inactive ISA2[47]. Similarly, in *Arabidopsis thaliana*, catalytically active ISA1 and non-catalytic subunit ISA2 form a heteromultimeric complex for interdependent stability[48]. Loss of DBE activity causes decreases in starch content, and instead, formation of highly branched water-soluble phytoglycogen[34–37]. Taken together, the results indicate that KOR1 accumulates carbohydrate as phytoglycogen most likely due to *ISA1* dysfunction, while *ISA1* deserves further examination by reverse genetics approaches in the future[12]. Amplified fluctuation of carbohydrate content (Fig. 1d) and loss of starch-like small granules after the dark periods (Fig. 3) further suggests that KOR1 carbohydrate is highly degradable. Carbohydrate fluctuation is also observed in the *C. reinhardtii* *sta7-10* mutant (Supplementary Fig. 3), indicating this is a universal phenotype in DBE-deficient microalgae.

KOR1 shows lower biomass relative to JSC4 after nitrate depletion (Fig. 1b). Growth suppression under nitrogen depletion has been reported in other starch-related mutant microalgae, including the DBE-deficient *C. reinhardtii* *sta7-10* mutant[28]. In glycogen-deficient cyanobacteria lacking in either AGPase (*glgC*) or glycogen synthase (*glgA₁* and *glgA₂*), decreases in cell viability under dark conditions were reported[49]. Since chlorophyll $a + b$ content does not decrease (Fig. 4c) and the de novo synthesis of 3PG increases (Fig. 6) in KOR1, $CO_2$ fixation is not assumed as the cause of decreased biomass. When comparing Δbiomass during light periods and dark periods, JSC4 and KOR1 are similar during 12 h light periods, while KOR1 biomass is significantly lower during 12 h dark periods (Supplementary Fig. 6). The above findings suggest that DBE deficiency promotes decreased biomass during dark periods. It is hypothesized that excretion of

carbohydrate degradation products, probably $CO_2$, might be the cause of decreased biomass in KOR1. In addition, culture volume-based production of lutein, β-carotene, and chlorophyll $a + b$ are almost same between JSC4 and KOR1 (Supplementary Fig. 7). This suggests that increases in pigment content during dark periods are the result of decreasing biomass. On the other hand, the transient increase in lutein content that occurred during light periods might be caused by increased levels of intermediate metabolites including DXP and MEcPP (Fig. 5). Similar to KOR1, increased carotenoid content is found in the DBE-deficient *C. reinhardtii* *sta7-10* (Supplementary Fig. 4) and reported in a starch-deficient *D. tertiolecta* mutant[30], suggesting that this is a universal phenotype of microalgae with blocked starch synthesis.

As biosynthesis of starch and lipid are interdependent based on shared intermediate metabolites[50], metabolome analysis is a rational approach to understand phenotypes of DBE-deficient mutants. This study provides evidence that DBE deficiency comprehensively increases intermediate metabolites related to lipid and carotenoid synthesis including G3P, pyruvate, acetyl-CoA, and DXP (Fig. 5), linking the metabolic precursors to lipid and carotenoid accumulation. In the AGPase-deficient *C. renhardtii* *sta6* mutant, intermediate metabolites involved in the synthesis of starch, lipids, and amino acids all accumulate[46]. In the cyanobacteria *Synechocystis* sp. PCC 6803 and *Synechococcus* sp. PCC 7002, impaired glycogen synthesis causes carbon redistribution leading to increased carbon flux in glycolysis and the TCA cycle, and also increases metabolites secreted in the culture supernatant (referred to as "metabolite overflow")[49,51–53]. Thus, increased intermediate metabolites might be a universal phenomenon in starch-deficient microalgae, including AGPase and DEB mutants, as well as glycogen-deficient cyanobacteria. The current study is the first to include dynamic metabolic profiling of starch synthesis-deficient microalgae by in vivo ¹³C labeling with $NaH^{13}CO_3$ as the carbon source. This labeling experiment shows increased de novo synthesis of G3P and PEP in KOR1 under illuminated condition (Fig. 6), revealing increased carbon flux from $CO_2$ toward lipid synthesis pathways. This result also supports the hypothesis that blocking of starch synthesis redirects carbon flux into lipid synthesis[25,44]. Increased de novo synthesis of F6P and G6P in starch synthesis/degradation pathways (Fig. 6) suggests that DBE-deficiency also accelerates synthesis of carbohydrate. Acceleration of both synthesis and degradation of carbohydrate in the DBE-deficient mutant presumably results from the increased surface-to-volume ratio of granules identified from the TEM images (Fig. 3) and water-solubility of phytoglycogen, relative to that of typical microalgae-insoluble starch granules.

The visualization of morphological features reveals the absence of large starch granules in KOR1 (Fig. 3), and this phenotype is consistent with previous reports on the DBE-deficient *C. reinhardtii* *sta7* mutant[34,35]. TEM analysis also reveals the existence of a large periplasmic space in KOR1 cells, which may result from the loss of the large starch granules. This means that the DEB-deficient microalgae possess additional cellular space, suggesting additional potential for further accumulation of lipid or other useful compounds. As discussed above, DBE-deficient microalgae possess many beneficial characteristics, and are easily obtained through selective breeding with FACS in a high-throughput manner[14,15,17–19]. In conclusion, this study emphasizes that disruption of DBE is a promising approach to improve production of lipid and other related compounds of value.

## Methods

**Strains and culture conditions.** *Chlamydomonas* sp. JSC4, isolated from Taiwanese brackish water, was used as the parental strain[20,21]. The 18S rRNA sequence of JSC4 has been deposited in the National Center for Biotechnology Information

GenBank with an accession number of KF383270. *Chlamydomonas* sp. KOR1 was newly obtained by the mutational breeding described below. To culture JSC4 and KOR1 photoautotrophically, modified bold (MB) 6 N medium ($8.82 \times 10^{-3}$ M NaNO$_3$, $2.20 \times 10^{-4}$ M K$_2$HPO$_4$, $3.04 \times 10^{-4}$ M MgSO$_4$, $6.47 \times 10^{-4}$ M KH$_2$PO$_4$, $4.28 \times 10^{-4}$ M NaCl, $1.71 \times 10^{-4}$ M CaCl$_2$, $6.55 \times 10^{-6}$ M FeCl$_3$, $2.55 \times 10^{-7}$ M ZnSO$_4$, $5.69 \times 10^{-8}$ M CoSO$_4$, $2.42 \times 10^{-6}$ M MnSO$_4$, $6.12 \times 10^{-9}$ M Na$_2$MoO$_4$, $9.99 \times 10^{-9}$ M Na$_2$SeO$_3$, and $6.26 \times 10^{-9}$ M NiCl$_2$) containing 2% (w/v) sea salt (Sigma-Aldrich Co., MO, USA) was used[10]. *C. reinhardtii* sta7-10 mutant (CC-5129) and its parental strain CC-425 were obtained from the Chlamydomonas Resource Center[33], and were cultured in TAP medium[54]. Microalgae cells were inoculated at an initial concentration of 20 mg g-DCW$^{-1}$ and cultured by using double-deck photobioreactors (upper stage containing 70 mL of medium and lower stage containing 50 mL of 2 M KHCO$_3$/K$_2$CO$_3$ which supplies 2% CO$_2$ gas to upper stage) under illumination of white fluorescent lamps (250 µmol photons m$^{-2}$ s$^{-1}$, 12 h:12 h light/dark cycling) at 30 °C with rotary shaking at 100 rpm[10].

**Breeding of lipid-rich mutants with light/dark selection.** For mutagenesis, JSC4 cells seeded on TAP agar plates[54] were irradiated with 50 Gy of carbon ion beams ($^{12}$C$^{5+}$, accelerated energy; 220 MeV, surface LET; 107 keV µm$^{-1}$) accelerated by an azimuthal varying field (AVF) cyclotron at Takasaki Ion Accelerators for Advanced Research Application (TIARA) of National Institutes for Quantum and Radiological Science and Technology (QST)[23,24]. For screening, mutagenized cells were cultured for 7 days under the light/dark cycling described above, and then intracellular lipid droplets were stained by 50 µM BODIPY 505/515 (4,4-Difluoro-1,3,5,7-Tetramethyl-4-Bora-3a,4a-Diaza-s-Indacene, Thermo Fisher Scientific, MA, USA) for 5 min[16,55]. Lipid-rich cells were sorted by an SH800 fluorescence-activated cell sorter (SONY, Tokyo, Japan) using BODIPY fluorescence and chlorophyll auto-fluorescence as the indicators of lipid and cell size, respectively[17–19]. The screening procedure of cultivation under the light/dark conditions and FACS-based sorting were repeated three times.

**Genomic analysis.** About 8 mL wet volume of KOR1 cells were harvested by centrifugation at 5000×*g* for 1 min. The cell pellet was washed once with distilled water, frozen in liquid nitrogen, and then milled using a mortar. Genomic DNA was purified from the frozen cell powder using NucleoBond Buffer Set IV and NucleoBond AXG 100 Column (MACHEREY-NAGEL, North Rhine-Westphalia, Germany) according to the manufacturer's manual. Library preparation and whole genome sequencing were performed by Takara Bio (Shiga, Japan). The sequence data of KOR1 were analyzed using CLC Genomics Workbench 12.0 (CLC bio, Aarhus, Denmark), and compared to the genomic sequence of JSC4 determined in the previous study[22].

**Zymography.** To prepare cell extract, microalgae cells at day 4.0 were suspended in extraction buffer (50 mM imidazole–HCl (pH = 7.4), 8 mM MgCl$_2$, 50 mM 2-mercaptoethanol, and 12.5% (v/v) glycerol), frozen and thawed with liquid nitrogen and 30 °C water four times, and then sonicated (output 4, duty 50%, 5 min by a ultrasonic disruptor UD-201; Tomy Seiko, Tokyo, Japan). The suspension was centrifuged at 20,000 × *g* for 20 min at 4 °C to obtain the supernatant[22]. For activity staining of DBE, native-PAGE was performed using the cell extract containing 10 µg of protein and an acrylamide gel containing 0.8% (w/v) potato amylopectin (A8515, Sigma-Aldrich Co.) with a constant current of 15 mA. After electrophoresis, the gel was incubated in reaction buffer (50 mM citric–Na$_2$HPO$_4$ (pH = 6.0) and 50 mM 2-mercaptoethanol) at 30 °C for 2 h. DBE activity was detected by staining the gel with 0.1% (w/v) I$_2$/1% (w/v) KI solution[32].

**Evaluation of biomass and nitrate concentration.** Biomass concentration was determined by weighing lyophilized cells harvested in microtubes. Cells were harvested by centrifugation at 5000 × *g* for 1 min, washed one with distilled water, and then lyophilized. Residual nitrate concentration was determined by measuring optical density of the culture supernatant at 220 nm using a UV mini-1240 UV–Vis spectrophotometer (Shimadzu, Kyoto, Japan)[10,56].

**Measurement of lipid.** Lyophilized cells prepared above were used for lipid measurement according to the previous study[10]. After adding heptadecanoic acid (Sigma-Aldrich Co.) as an internal standard, lyophilized cells were fractured with 0.5 mm glass beads YGB05 and a multi-bead shocker MB1001C(S) (Yasui Kikai, Osaka, Japan) at 4 °C. Lipids were esterified using a Fatty Acid Methylation Kit (Nacalai Tesque, Kyoto, Japan), and then the fatty acid methyl esters were identified and quantified using a GCMS-QP2010 Plus (Shimadzu) and a capillary column DB-23 (0.15 µm, 60 m × 0.25 mm; Agilent Technologies, CA, USA).

**Measurement of carbohydrate.** Lyophilized cells prepared above were used for carbohydrate measurement using hot acid hydrolysis according to the previous study[10]. Briefly, 3 mg of lyophilized cells were suspended in 2 mL of 4% (v/v) H$_2$SO$_4$, autoclaved at 120 °C for 30 min, neutralized by adding 1 mL of 22% (w/v) Na$_2$CO$_3$, and then cell debris was removed by filtrating with a Shim-pack SPR-Pb column (Shimadzu). Soluble starch (CAS number: 9005-84-9, Nacalai Tesque, Kyoto, Japan) was used as a quantitative standard. The glucose concentration was

measured using an HPLC system (Shimadzu) and an Aminex HPX-87H column (9 µm, 300 mm × 7.8 mm; Bio-Rad Laboratories, CA, USA).

**Transmission electron microscopy.** Cells were harvested by centrifugation at 5000 × *g* for 1 min, and then immediately subjected to chemical fixation as follows. Cells were fixed with 50 mM cacodylate buffer (pH = 7.4) containing 2% paraformaldehyde and 2% glutaraldehyde at 4 °C overnight, washed three times with the same buffer, and postfixed with 50 mM cacodylate buffer (pH = 7.4) containing 2% osmium tetroxide at 4 °C for 3 h. The succeeding sample processing was performed by Tokai Electron Microscopy (Aichi, Japan). After ultra-thin sectioning and mounting on copper grids, samples were successively stained with 2% uranyl acetate and lead stain solution (Sigma-Aldrich Co.), and then observed with a transmission electron microscope JEM-1400Plus (JEOL Ltd., Tokyo, Japan).

**Measurement of pigments.** Lyophilized cells prepared above were used for measurement of carotenoids and chlorophylls. 3 mg of lyophilized cells was suspended in 500 µL of pre-cooled (4 °C) methanol:acetone = 1:1 (v/v), and fractured with 0.5 mm glass beads YGB05 and a multi-bead shocker MB1001C(S) (Yasui Kikai) at 4 °C. After centrifugation at 10,000×*g* for 5 min, 150 µL of the supernatant was dried under vacuum using a centrifugal evaporator CEV-3100 (EYELA, Tokyo, Japan). The dried sample was resuspended in 500 µL of acetonitrile: chloroform = 8:2 (v/v) containing 1 µM *trans*-β-apo-8′-carotenal as an internal standard, and filtered by a 0.22 µm Cosmospin Filter G (Nacalai Tesque, Kyoto, Japan). Then samples were subjected to identification and quantification of pigments with an ACQUITY ultra performance liquid chromatography system equipped with a photodiode array detector and a BEH Shield RP18 column (1.7 µm, 2.1 mm × 100 mm; Waters, MA, USA)[57].

**Metabolome analysis.** Sample preparation for metabolome analysis was conducted as described in the previous studies with modifications[10,22]. To prepare intracellular metabolites, cultured cells equivalent to 5 mg-DCW were harvested with 10.0 µm pore size polytetrafluoroethylene filters (Merck Millipore, Burlington, MA, USA), washed with distilled water, and immediately suspended into 1 mL of pre-cooled (−30 °C) methanol containing 36 µM piperazine-1,4-bis(2-ethane-sulfonic acid) (Dojindo Laboratories, Kumamoto, Japan) as an internal standard. 500 µL of the cell suspension was fractured with 0.5 mm glass beads YGB05 and a multi-bead shocker MB1001C(S) (Yasui Kikai) at 4 °C, and then mixed with 150 µL of chloroform and 50 µL of ultrapure water. 400 µL of the supernatant after centrifugation at 14,000 × *g* for 5 min was transferred to a clean tube, and then 200 µL of ultrapure water was added. After centrifugation at 14,000×*g* for 5 min, the aqueous layer was filtered by an Amicon Ultra-0.5 Centrifugal Filter Unit UFC5003BK (Merck Millipore). 300 µL of flow-through was dried under vacuum using a centrifugal evaporator CEV-3100 (EYELA, Tokyo, Japan), re-dissolved in 20 µL of ultrapure water, and then subjected to capillary electrophoresis–mass spectrometry (CE–MS) using G7100 CE and G6224AA LC/MSD time-of-flight systems (Agilent Technologies).

To evaluate the newly synthesized metabolites, in vivo $^{13}$C labeling was performed using NaH$^{13}$CO$_3$ as a carbon source[10,22]. Briefly, cultured cells at day 5.5 were harvested with 10.0 µm pore size polytetrafluoroethylene filters (Merck Millipore), washed with distilled water, and resuspended in MB 6 N medium containing 2% (w/v) sea salt and 25 mM NaH$^{13}$CO$_3$ (Cambridge Isotope Laboratories, Inc., Tewksbury, MA, USA). After labeling for 0, 2.5, 5, 10 min under illumination of white fluorescent lamps at 250 µmol photons m$^{-2}$ s$^{-1}$ with stirring at 100 rpm, intracellular metabolites were prepared and analyzed as described above. $^{13}$C labeling was calculated by multiplying pool size and $^{13}$C fraction, which is the ratio of $^{13}$C in total carbon determined by mass shifts from the $^{12}$C to $^{13}$C mass spectra[43].

**Statistics and reproducibility.** Data in this study are represented as mean ± standard deviation of three replicate experiments. Statistical significance was determined by Welch's *t* test.

**Reporting summary.** Further information on research design is available in the Nature Research Reporting Summary linked to this article.

## Data availability

The source data underlying Figs. 1, 4, 5, and 6 are provided in Supplementary Data 1. The sequencing data are deposited in the DNA Data Bank of Japan (DDBJ, https://www.ddbj.nig.ac.jp) with the accession code DRA011641. The data supporting the findings of this study are also available from the corresponding author upon reasonable request.

## Code availability

All codes are available from the corresponding author upon reasonable request.

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

## Acknowledgements

We would like to thank Dr. Masao Mochizuki, Ms. Shiori Funaki, Ms. Aya Narita, and Ms. Yuko Yoshikawa for technical assistance. This study was financially supported by the ImPACT Program of Council for Science, Technology and Innovation (Cabinet Office, Government of Japan), and the Adaptable and Seamless Technology transfer Program through Target-driven R&D (A-STEP) from Japan Science and Technology Agency (JST).

## Author contributions

Y.K. designed the study, conducted the experiments, and drafted the manuscript. T.O. and K.I. performed mutation analysis and revise the manuscript. C.J.V. helped interpret results and revise the manuscript. M.M. performed metabolome analysis. R.H. helped interpret results. K.S. and Y.O. performed mutagenesis and revised the manuscript. J.-S. C. provided strain JSC4 and commented on the study. T.H. designed the study, revised the manuscript, and supervised the study. A.K. commented on the study and assisted with the laboratory management. All authors read and approved the final version of the manuscript.

## Competing interests

The authors declare no competing interests.
