## [Peer Review File · Communications Biology]

Reviewers' comments:

Reviewer #1 (Remarks to the Author):

The authors describe the isolation of a mutant (KOR1) in the green algae *Chlamydomonas* that accumulates lipid even during diurnal cycles. They isolated this mutant by using carbon ion beam mutagenesis followed by FACS sorting of high lipid cells grown in a diurnal cycle. While many studies have used FACS to isolate high lipid mutants, none have focused on those that do so during diurnal growth which would be requisite for biofuel production. Sequencing this mutant revealed a disruption in the Isoamylase 1 (Isa1) debranching enzyme (DBE). Mutants in this gene have been extensively characterized for their high lipid phenotype. However, no studies I am aware of have looked into how this mutation impacts lipid accumulation during diurnal cycles and the authors generate significant physiological and metabolomic data about this mutant during a diurnal cycle. This work will be of interest to others in the field and this study advances our understanding of how disruption of Isa1 impacts the alga and imparts a high lipid phenotype.

I find the data to appear very solid and the conclusions drawn largely justified. However, there are a few areas that the authors could look into:

194 – “KOR1 lutein and β -carotene fluctuate in response to light/dark cycling with decreases during the light periods and increases during the dark periods, suggesting a relationship to lipid accumulation.”

Could the fluctuations in dry cell weight (Fig 1) be driving the fluctuations seen in lutein/beta carotene (Fig4) since this is on a dry cell weight basis? Are lutein and beta-carotene cycling the same way on a per cell basis during this time period? Looking into the constant light data it seems to be about the same magnitude of change if you do not take into the 'spike' in concentration during the dark period. If the lutein and beta carotene are truly cycling, is it known that carotenoids are catabolized in the light? It would seem like this would they be catabolized during the light?

“Chlorophyll a+b content does not dramatically change in KOR1, although there is a temporal increase during light periods (Fig. 4c).” The data looks as if there is only an increase in KOR1 chlorophyll during N replete conditions and it is increasing in both light and dark. In the N deplete times it appears to be the opposite of the author's statement, decreasing during the light. However, the apparent 'increase' during the night could be due to the DCW cycling, the same as above.

Line 330 “critically disrupted DBE activity (Fig. 2)” Fig 2 does not show directly DBE activity is abolished, only that the Isa1 gene is disrupted. To directly show that DBE activity has been eliminated a zymogram assay can be performed.

Additional points the authors could consider:

The discussion is a bit repetitive. Some sections could be combined to make it more concise and streamlined.

Did the authors ever investigate that other *Chlamydomonas* Isa1 mutants (sta7) have a similar phenotype in a diurnal cycle, or is this a phenotype dependent on the JSC4 background.

Could the decrease in biomass in the dark be a result of increased respiration? If the phytyglycogen is more easily degradable than starch, is more respired (and thus loss via CO₂)?

Were any other mutations detected in KOR1 besides the ones found in Isa1?

The method used to measure carbohydrates can distinguish between glucose and phytyglycogen. Could an increase in carbohydrates not found in phytyglycogen be responsible for some of the phenotype observed (such as soluble sugars)?

384 – “¹³CO₂ as the carbon source.” Or is it “NaH¹³CO₃ as a carbon source” in line 507.

395 – “sta7-10 mutant ref32,33” These references are to different sta7, not sta7-10; sta7-10 was isolated in Posewitz 2004 Plant Cell, Hydrogen Photoproduction Is Attenuated by Disruption of an Isoamylase Gene in *Chlamydomonas reinhardtii*

Reviewer #2 (Remarks to the Author):

The authors are reporting altered physiology effects and a set of key metabolic parameters of *Chlamydomonas* sp. KOR1, a highly interesting new mutant accumulating lipids and carotenoids in dark/ light cultivations. This mutant was generated by unspecific irradiation and several rounds of breeding. The breeding and selection target was an increased lipid and carotenoid content after cultivation in alternating light and dark conditions. Alternating light and dark phases are very important for the productivity of outdoor bioprocesses based on phototrophic microorganisms. Cultivations of microalgae obviously are characterized by these day and night (light and dark) cycles of sunlight. Lipid and carotenoid overproducing microalgae are described in literature, yet they were not adapted towards alternating light and dark cycles. The parental strain of the KOR1 mutant for example, *Chlamydomonas* sp. JSC4, accumulates a lot of lipids under nitrogen depletion and salinity stress conditions, yet it is only characterized under light cultivation and shows decreased lipid accumulation during night and dark cycles.

The manuscript reports a detailed set of physiological and metabolome data of the newly generated mutant KOR1. These data are well presented and highly interesting for understanding and for characterization of the mutant KOR1. As a reviewer, at this point, I can unfortunately not go into the details of these sets of physiological data. The main claim of the authors is that the ISA1 gene, encoding a starch debranching enzyme (DBE), is responsible for the physiological changes observed in the mutant KOR1 relative to the parental strain *Chlamydomonas* sp. JSC4 and that its disruption can improve biofuel production. Carbon ion beam mutagenesis was used for constructing the KOR1 strain. Despite the fact, that no mutations were found in 15 other genes encoding starch metabolising enzymes, it is obvious from the mutation frequency in the genomic area of the ISA1 gene (Figure 2), that the rest of the genome of the parental strain *Chlamydomonas* sp. JSC4 is also highly mutated. It is very unlikely that all these mutations are silent mutations (as the mutations in the ISA1 gene are expected to be deleterious). It is thus not possible to claim all or even part of the physiological changes in the KOR1 mutant over the parental strain JSC4 to the ISA1 gene (DBE).

One possible test of the argumentation of the authors that the ISA1 gene is indeed the main target for strain engineering and also a reason for the high lipid production and accumulation features of KOR1 would be a targeted deletion of the ISA1 gene in the parental strain *Chlamydomonas* species JSC4. Obviously this is a major effort, but without such a control experiment the specific claim that the ISA1 gene is responsible for the physiological changes observed is not strong.

If such an experiment shows that some key physiological parameters are the same as observed in the KOR1 mutant, a specific reviewing of the physiological data sets can be done. In case the authors already have conclusive other genome analyses or experimental data supporting their main claim of the ISA1 gene being the main engineering target and cause of the observed physiological effects, than this has to be included in the present manuscript and I will be glad to further review this important paper.

Overall the topic addressed by the authors is of key importance to industrial applications of microalgae and the physiological data and also the regulation circuit presented in Figure 7 are highly interesting.

KOBE UNIVERSITY

Tomohisa Hasunuma
Engineering Biology Research Center
1-1 Rokkodai, Nada, Kobe 657-8501, Japan
Tel. +81-78-803-6202 Fax. +81-78-803-6102
E-mail: hasunuma@port.kobe-u.ac.jp

Responses to the reviewers' comments

We thank the reviewers for taking time to review our manuscript and for giving us valuable comments. Our point-to-point responses to the comments are as follows. The changes in the revised manuscript are shown as red text.

Reviewers' comments:

Reviewer #1 (Remarks to the Author):

The authors describe the isolation of a mutant (KOR1) in the green algae Chlamydomonas that accumulates lipid even during diurnal cycles. They isolated this mutant by using carbon ion beam mutagenesis followed by FACS sorting of high lipid cells grown in a diurnal cycle. While many studies have used FACS to isolate high lipid mutants, none have focused on those that do so during diurnal growth which would be requisite for biofuel production. Sequencing this mutant revealed a disruption in the Isoamylase 1 (Isa1) debranching enzyme (DBE). Mutants in this gene have been extensively characterized for their high lipid phenotype. However, no studies I am aware of have looked into how this mutation impacts lipid accumulation during diurnal cycles and the authors generate significant physiological and metabolomic data about this mutant during a diurnal cycle. This work will be of interest to others in the field and this study advances our understanding of how disruption of Isa1 impacts the alga and imparts a high lipid phenotype.

I find the data to appear very solid and the conclusions drawn largely justified. However, there are a few areas that the authors could look into:

194 – “KOR1 lutein and β -carotene fluctuate in response to light/dark cycling with decreases during the light periods and increases during the dark periods, suggesting a relationship to lipid accumulation.”

Could the fluctuations in dry cell weight (Fig 1) be driving the fluctuations seen in lutein/beta carotene (Fig4) since this is on a dry cell weight basis? Are lutein and beta-carotene cycling the same way on a per cell basis during this time period? Looking into the constant light data it seems to be about the same magnitude of change if you do not take into the ‘spike’ in concentration during the dark period. If the lutein and beta carotene are truly cycling, is it known that carotenoids are catabolized in the light? It would seem like this would they be catabolized during the light?

KOBE UNIVERSITY

Tomohisa Hasunuma
Engineering Biology Research Center
1-1 Rokkodai, Nada, Kobe 657-8501, Japan
Tel. +81-78-803-6202 Fax. +81-78-803-6102
E-mail: hasunuma@port.kobe-u.ac.jp

Response: The positive comments on our manuscript are really appreciated. To address the first comment:

(1) We now consider that the fluctuations in dry cell weight are driving the fluctuations seen in lutein and β -carotene contents on a dry cell weight basis during light/dark cycling. In order to examine this, we checked culture volume-based production of these carotenoids, independent from dry cell weight. The volume-based result shows that amount of these carotenoids in KOR1 are actually not fluctuating during light/dark cycling, and are almost same between JSC4 and KOR1. Thanks to this insightful comment we now can include this necessary data in the discussion of carotenoid content. Accordingly, the culture volume-based data (Supplementary Figure 5) and discussion of volume-based production were added to the manuscript.

(2) We are not sure if lutein and β -carotene are cycling in the same way on a per cell basis. To examine this, measurement of cell number per culture volume is needed. Unfortunately, cell number cannot be measured accurately because *Chlamydomonas* cells form aggregates in the sea salt-containing medium used in this study.

“Chlorophyll a+b content does not dramatically change in KOR1, although there is a temporal increase during light periods (Fig. 4c).” The data looks as if there is only an increase in KOR1 chlorophyll during N replete conditions and it is increasing in both light and dark. In the N deplete times it appears to be the opposite of the author’s statement, decreasing during the light. However, the apparent ‘increase’ during the night could be due to the DCW cycling, the same as above.

Response: We thank the reviewer for this comment.

(1) We apologize for making the following mistake: *“In the N deplete times it appears to be the opposite of the author’s statement, decreasing during the light.”* The temporal increases in chlorophyll actually occurred during dark periods. In addition, we should describe whether we are referring to N replete conditions or N deplete conditions. Therefore, this sentence has been revised appropriately.

(2) This important comment suggests that the increase in chlorophyll is due to cycling of dry cell weight. As performed for the first comment, we also checked culture volume-based production of chlorophyll. This result shows that the amount of chlorophyll in KOR1 culture does not fluctuate during light/dark cycling, suggesting that the increase in chlorophyll during the night is due to dry cell weight cycling. The culture volume-based data for chlorophyll (Supplementary Figure 5) and an explanation have also been added to the revised manuscript.

Line 330 “critically disrupted DBE activity (Fig. 2)” Fig 2 does not show directly DBE activity is abolished, only that the Isal gene is disrupted. To directly show that DBE activity has been

KOBE UNIVERSITY

Tomohisa Hasunuma
Engineering Biology Research Center
1-1 Rokkodai, Nada, Kobe 657-8501, Japan
Tel. +81-78-803-6202 Fax. +81-78-803-6102
E-mail: hasunuma@port.kobe-u.ac.jp

eliminated a zymogram assay can be performed.

Response: As the reviewer pointed out, this was a misleading and incorrect statement. We additionally performed a zymography experiment according to this comment, and found that activity toward amylopectin is significantly depressed in KOR1 compared to that of JSC4. This must be important data as it directly shows that DBE activity in KOR1 is significantly abolished. Therefore, this zymography data is added as Supplementary Figure 2 and descriptions of this experiment are included in the revised results and discussion sections. Thank you for the valuable comment.

Additional points the authors could consider:

The discussion is a bit repetitive. Some sections could be combined to make it more concise and streamlined.

Response: We apologize for the repetitive discussion. According to this comment, descriptions in the discussion section were arranged.

*Did the authors ever investigate that other *Chlamydomonas* *Isa1* mutants (*sta7*) have a similar phenotype in a diurnal cycle, or is this a phenotype dependent on the JSC4 background.*

Response: We thank the reviewer for this important comment. In response, we investigated the phenotype of *Chlamydomonas reinhardtii sta7-10* mutant under light/dark cycling to determine if the KOR1 phenotypes are limited to microalgae that have the JSC4 background. Similar to KOR1, the *sta7-10* mutant showed the same patterns of fluctuations in the accumulation of carbohydrate, lipid, and pigments. This is strong evidence to support that KOR1 phenotypes are indeed caused by DBE deficiency and are not dependent on the JSC4 background. Therefore, these important results were added as Supplementary Figure 3 and 4, and descriptions of these experiments were also added to the results and discussion sections.

Could the decrease in biomass in the dark be a result of increased respiration? If the phytyglycogen is more easily degradable than starch, is more respired (and thus loss via CO₂)?

Response: It is uncertain whether the decrease of biomass in the dark is a result of increased respiration. We hypothesize that excretion of phytyglycogen degradation products may be the cause of decreased biomass in KOR1 during dark periods. Although we have not yet investigated what the excreted molecules are, this must be an important point. Therefore, a description of this was added

KOBE UNIVERSITY

Tomohisa Hasunuma
Engineering Biology Research Center
1-1 Rokkodai, Nada, Kobe 657-8501, Japan
Tel. +81-78-803-6202 Fax. +81-78-803-6102
E-mail: hasunuma@port.kobe-u.ac.jp

to the discussion section.

Were any other mutations detected in KOR1 besides the ones found in Isa1?

Response: Many mutations were detected in KOR1, and most are single nucleotide variants. Because single nucleotide variants are generated as spontaneous mutations during normal microalgae cultivation, most were not considered as the most probable causative mutations, unless they resulted in significant amino acid changes. Large insertion/deletion mutations, that have the highest impact on the cell and are often induced by ion beam irradiation, were considered as noteworthy candidates. However, such large mutations were not detected in KOR1, with exception to that in the *ISA1* gene.

The method used to measure carbohydrates can distinguish between glucose and phytyglycogen. Could an increase in carbohydrates not found in phytyglycogen be responsible for some of the phenotype observed (such as soluble sugars)?

Response: The reviewer's suggestion is very interesting. An increase in non-phytyglycogen carbohydrates such as soluble sugars might be responsible for some of the KOR1 phenotype. Unfortunately, we did not analyze glucose and phytyglycogen separately, and it is uncertain how much of the measured glucose is present as soluble sugar in the cells.

384 – “ $^{13}\text{CO}_2$ as the carbon source.” Or is it “ $\text{NaH}^{13}\text{CO}_3$ as a carbon source” in line 507.

Response: We apologize for our mistake and thank the reviewer for carefully pointed this out. This description was corrected as “ $\text{NaH}^{13}\text{CO}_3$ as the carbon source.”

395 – “sta7-10 mutant ref32,33” These references are to different sta7, not sta7-10; sta7-10 was isolated in Posewitz 2004 Plant Cell, Hydrogen Photoproduction Is Attenuated by Disruption of an Isoamylase Gene in Chlamydomonas reinhardtii

Response: Thanks again to the reviewer for pointing out this mistake. We have corrected the descriptions appropriately.

KOBE UNIVERSITY

Tomohisa Hasunuma
Engineering Biology Research Center
1-1 Rokkodai, Nada, Kobe 657-8501, Japan
Tel. +81-78-803-6202 Fax. +81-78-803-6102
E-mail: hasunuma@port.kobe-u.ac.jp

Reviewer #2 (Remarks to the Author):

The authors are reporting altered physiology effects and a set of key metabolic parameters of Chlamydomonas sp. KOR1, a highly interesting new mutant accumulating lipids and carotenoids in dark/ light cultivations. This mutant was generated by unspecific irradiation and several rounds of breeding. The breeding and selection target was an increased lipid and carotenoid content after cultivation in alternating light and dark conditions. Alternating light and dark phases are very important for the productivity of outdoor bioprocesses based on phototrophic microorganisms. Cultivations of microalgae obviously are characterized by these day and night (light and dark) cycles of sunlight. Lipid and carotenoid overproducing microalgae are described in literature, yet they were not adapted towards alternating light and dark cycles. The parental strain of the KOR1 mutant for example, Chlamydomonas sp. JSC4, accumulates a lot of lipids under nitrogen depletion and salinity stress conditions, yet it is only characterized under light cultivation and shows decreased lipid accumulation during night and dark cycles.

The manuscript reports a detailed set of physiological and metabolome data of the newly generated mutant KOR1. These data are well presented and highly interesting for understanding and for characterization of the mutant KOR1. As a reviewer, at this point, I can unfortunately not go into the details of these sets of physiological data. The main claim of the authors is that the ISAI gene, encoding a starch debranching enzyme (DBE), is responsible for the physiological changes observed in the mutant KOR1 relative to the parental strain Chlamydomonas sp. JSC4 and that its disruption can improve biofuel production. Carbon ion beam mutagenesis was used for constructing the KOR1 strain. Despite the fact, that no mutations were found in 15 other genes encoding starch metabolising enzymes, it is obvious from the mutation frequency in the genomic area of the ISAI gene (Figure 2), that the rest of the genome of the parental strain Chlamydomonas sp. JSC4 is also highly mutated. It is very unlikely that all these mutations are silent mutations (as the mutations in the ISAI gene are expected to be deleterious). It is thus not possible to claim all or even part of the physiological changes in the KOR1 mutant over the parental strain JSC4 to the ISAI gene (DBE).

One possible test of the argumentation of the authors that the ISAI gene is indeed the main target for strain engineering and also a reason for the high lipid production and accumulation features of KOR1 would be a targeted deletion of the ISAI gene in the parental strain Chlamydomonas species JSC4. Obviously this is a major effort, but without such a control experiment the specific claim that the ISAI gene is responsible for the physiological changes observed is not strong.

If such an experiment shows that some key physiological parameters are the same as observed in the KOR1 mutant, a specific reviewing of the physiological data sets can be done. In case the

KOBE UNIVERSITY

Tomohisa Hasunuma
Engineering Biology Research Center
1-1 Rokkodai, Nada, Kobe 657-8501, Japan
Tel. +81-78-803-6202 Fax. +81-78-803-6102
E-mail: hasunuma@port.kobe-u.ac.jp

authors already have conclusive other genome analyses or experimental data supporting their main claim of the ISAI gene being the main engineering target and cause of the observed physiological effects, than this has to be included in the present manuscript and I will be glad to further review this important paper.

Overall the topic addressed by the authors is of key importance to industrial applications of microalgae and the physiological data and also the regulation circuit presented in Figure 7 are highly interesting.

Response: We are grateful to the reviewer for their interest in our study and for providing us with critical comments.

(1) According to the comment: “it is obvious from the mutation frequency in the genomic area of the ISAI gene (Figure 2), that the rest of the genome of the parental strain *Chlamydomonas* sp. JSC4 is also highly mutated”, this suggests that there may be many non-silent and deleterious mutations in the KOR1 genome. We cannot deny the possibility that KOR1 possesses significant mutations other than that in the ISAI gene. However, ion beams generally cause a few local and serious mutations, and do not usually cause many small mutations uniformly throughout the genome like that caused by UV and chemical mutagens. Most importantly, we added phenotype investigation and characterization of the *Chlamydomonas reinhardtii* *sta7-10* mutant (CC-5129), which was generated by insertional transformation of the parental strain CC-425 and reported as a DBE mutant in a previous study. We found that CC-5129 showed fluctuations in accumulation of carbohydrate, lipid, and pigments during light/dark cycling, highly similar to that of KOR1. This result supports that the major causative mutation in KOR1 is that in ISAI gene alone. Therefore, this important new data was added as Supplementary Figure 3 and 4, and descriptions for these experiments were also added in the revised results and discussion sections.

(2) The reviewer also suggests the targeted deletion of the ISAI gene in *Chlamydomonas* sp. JSC4 as a strain engineering experiment to confirm the role of ISAI. We agree that it is necessary to carefully examine the causative mutation in KOR1 by reverse genetics approaches. Unfortunately, neither transgenic tools nor genome editing methods have been established in the *Chlamydomonas* sp. used in this study. We are now trying to establish these techniques, but the attempts have not been successful yet and it will require further time. To thoroughly address this comment and support the main claim of this study, we alternatively performed a validation experiment using the *Chlamydomonas reinhardtii* *sta7-10* mutant (CC-5129) as described above. We believe this result provides convincing evidence to support that the KOR1 phenotypes are caused by DBE deficiency. In addition, we performed a zymography experiment and found that DBE activity in KOR1 is highly depressed compared to JSC4 (added as Supplementary Figure 2). This further supports that ISAI plays a major role in DBE activity in the *Chlamydomonas* sp. used in this study.

KOBE UNIVERSITY

Tomohisa Hasunuma

Engineering Biology Research Center

1-1 Rokkodai, Nada, Kobe 657-8501, Japan

Tel. +81-78-803-6202 Fax. +81-78-803-6102

E-mail: hasunuma@port.kobe-u.ac.jp

REVIEWERS' COMMENTS:

Reviewer #1 (Remarks to the Author):

The authors have added additional experiments (zymogram, sta7-10 data) that make the manuscript stronger and further support their claims.

However, a few items still need to be addressed:

In finding that volume-based result shows that amount of these carotenoids in KOR1 are actually not fluctuating during light/dark cycling then the following sentence should be revised:

197 "KOR1 lutein and β -carotene fluctuate in response to light/dark cycling with decreases during the light periods and increases during the dark periods, suggesting a relationship to lipid and carbohydrate accumulation."

The addition of the culture volume-based data is a nice addition and the authors say it should be in Supplementary Figure 5 however all of the plots in that figure are based on g-DCW.

"The culture volume-based data for chlorophyll (Supplementary Figure 5)": Again, the only data I see in Supplementary Figure 5 for chlorophyll is mg/g-DCW.

Reviewer #2 (Remarks to the Author):

Thank you very much for adding additional information on the day/ night accumulation of lipids and carotenoid in the mutant *Chlamydomonas reinhardtii* sta7-10 (CC-5129). This addresses and positively answers my concern of background mutations in KOR1 resulting in the altered phenotype of this mutant.

KOBE UNIVERSITY

Tomohisa Hasunuma
Engineering Biology Research Center
1-1 Rokkodai, Nada, Kobe 657-8501, Japan
Tel. +81-78-803-6202 Fax. +81-78-803-6102
E-mail: hasunuma@port.kobe-u.ac.jp

Responses to the reviewers' comments

We are grateful to the reviewers for taking time to review our manuscript again. Our point-to-point responses to the comments are as follows.

Reviewers' comments:

Reviewer #1 (Remarks to the Author):

The authors have added additional experiments (zymogram, sta7-10 data) that make the manuscript stronger and further support their claims.

However, a few items still need to be addressed:

In finding that volume-based result shows that amount of these carotenoids in KOR1 are actually not fluctuating during light/dark cycling then the following sentence should be revised:

197 "KOR1 lutein and β -carotene fluctuate in response to light/dark cycling with decreases during the light periods and increases during the dark periods, suggesting a relationship to lipid and carbohydrate accumulation."

Response: Thank you for pointing out this incorrect statement. The sentence has been revised as follows: "DCW-based contents of KOR1 lutein and β -carotene fluctuate in response to light/dark cycling...".

The addition of the culture volume-based data is a nice addition and the authors say it should be in Supplementary Figure 5 however all of the plots in that figure are based on g-DCW.

"The culture volume-based data for chlorophyll (Supplementary Figure 5)": Again, the only data I see in Supplementary Figure 5 for chlorophyll is mg/g-DCW.

Response: We apologize for this mistake in our response file. In the revised manuscript, the culture volume-based data is now cited as Supplementary Fig. 7.

Reviewer #2 (Remarks to the Author):

*Thank you very much for adding additional information on the day/ night accumulation of lipids and carotenoid in the mutant *Chlamydomonas reinhardtii* sta7-10 (CC-5129). This addresses and*

KOBE UNIVERSITY

Tomohisa Hasunuma
Engineering Biology Research Center
1-1 Rokkodai, Nada, Kobe 657-8501, Japan
Tel. +81-78-803-6202 Fax. +81-78-803-6102
E-mail: hasunuma@port.kobe-u.ac.jp

positively answers my concern of background mutations in KOR1 resulting in the altered phenotype of this mutant.

Response: We thank the reviewer for this comment, and for the positive review.